# Efficacy of the Cardiac Implantable Electronic Device Multisensory Triage-HF Algorithm in Heart Failure Care: A Real-World Clinical Experience

**DOI:** 10.3390/s24113664

**Published:** 2024-06-05

**Authors:** Ugur Aslan, Saskia L. M. A. Beeres, Michelle Feijen, Gerlinde M. Mulder, J. Wouter Jukema, Anastasia D. Egorova

**Affiliations:** 1Department of Cardiology, Leiden University Medical Center, Albinusdreef 2, 2333 ZA Leiden, The Netherlands; u.aslan@lumc.nl (U.A.); s.l.m.a.beeres@lumc.nl (S.L.M.A.B.); g.m.mulder@lumc.nl (G.M.M.); j.w.jukema@lumc.nl (J.W.J.); 2Netherlands Heart Institute, Morseelsepark 1, 3511 EP Utrecht, The Netherlands

**Keywords:** cardiac implantable electronic device, multisensory algorithm, heart failure, telemonitoring, remote monitoring

## Abstract

Heart failure (HF) admissions are burdensome, and the mainstay of prevention is the timely detection of impending fluid retention, creating a window for medical treatment intensification. This study evaluated the accuracy and performance of a Triage-HF-guided carepath in real-world ambulatory HF patients in daily clinical practice. In this prospective, observational study, 92 adult HF patients (71 males (78%), with a median age of 69 [IQR 59–75] years) with the Triage-HF algorithm activated in their cardiac implantable electronic devices (CIEDs), were monitored. Following high-risk alerts, an HF nurse contacted patients to identify signs and symptoms of fluid retention. The sensitivity and specificity were 83% and 97%, respectively. The positive predictive value was 89%, and negative predictive value was 94%. The unexplained alert rate was 0.05 alerts/patient year, and the false negative rate was 0.11 alerts/patient year. Ambulatory diuretics were initiated or escalated in 77% of high-risk alert episodes. In 23% (n = 6), admission was ultimately required. The median alert handling time was 2 days. Fifty-eight percent (n = 18) of high-risk alerts were classified as true positives in the first week, followed by 29% in the second–third weeks (n = 9), and 13% (n = 4) in the fourth–sixth weeks. Common sensory triggers included an elevated night ventricular rate (84%), OptiVol (71%), and reduced patient activity (71%). The CIED-based Triage-HF algorithm-driven carepath enables the timely detection of impending fluid retention in a contemporary ambulatory setting, providing an opportunity for clinical action.

## 1. Introduction

Unplanned heart failure (HF) hospitalizations impose a substantial burden on the quality of patients’ lives and are associated with a poor prognosis [1,2,3,4,5,6]. Timely detection of congestion, which is key in preventing hospitalizations, remains challenging due to late symptom manifestation [7]. In the majority of patients admitted for decompensated HF, the emergency department is their initial place of contact even though there is accumulating evidence that detectable pathophysiological changes occur in the preceding weeks [8]. If these initial signs of impeding fluid retention were to be identified by telemonitoring at an earlier stage, a time window for treatment intensification in the home setting could be created to avert the need for hospitalization. Established telemonitoring strategies include the use of invasive hemodynamic sensors, like the CardioMEMS^TM^ system, enabling daily measurements of pulmonary pressures, which are known to increase several weeks before symptoms become apparent [9].

Conceptually, cardiac implantable electronic devices (CIEDs) provide a unique opportunity for telemonitoring as they have a large variety of sensors that can provide insight into the clinical status of a heart failure patient. The ability to continuously track (patho-)physiological trends over time without any extra effort from the patient makes CIEDs particularly attractive for this purpose [10,11,12,13,14,15,16,17]. In the previous decade, CIED-based monitoring strategies predominantly used the data of one single sensor. For example, OptiVolTM and CorVueTM aimed to detect fluid overload based on intrathoracic impedance alone. These single-sensor techniques, however, were demonstrated to not be robust enough in the timely detection of impending decompensation [10,18]. More recent strategies integrate data from multiple CIED-based sensors, computing the composite deviations into an HF risk index or alert status, giving an indication of the risk of an upcoming episode of fluid retention in the coming weeks–months [19,20,21]. Consequently, a few commercially available CIED-based multisensory algorithms have recently been developed [19,20,21].

One of these is the Triage-HF risk score algorithm (Medtronic, Minnesota, United States) that is designed to stratify patients as being at low, medium, or high risk for an HF event in the next 30 days by integrating specific physiological parameters such as thoracic impedance, arrhythmia burden, ventricular pacing percentage, night ventricular heart rate, heart rate variability, and patient activity levels [12,21,22,23].

The initial validation study of the Triage-HF score reported HF hospitalization rates of 0.6% for low-risk, 1.3% for medium-risk, and 6.8% for high-risk alerts within the next 30 days. This corresponds to a 10-fold higher risk of HF hospitalization with the “high-risk” alert status in the preceding 30 days and a 2.1-fold higher risk with the “medium-risk” alert status compared with the “low-risk” alert status [12]. This stratification highlights other research findings focusing on “high-risk score” alerts, indicating that only the high-risk status is associated with an increased risk of HF hospitalization, designating high-risk alerts as actionable. However, previous studies on the triage-HF risk algorithm’s performance have reported wide variability in its sensitivity, specificity, and positive predictive values in predicting HF hospitalizations among high-risk scores [12,22,23,24,25,26]. Its sensitivity ranged from 31.5% to 98.6%, its specificity from 63.4% to 90.1%, and its positive predictive values from 4.1% to 55.9%. This variability in performance can, at least partly, be explained by marked differences in the definitions of HF-related events and variations in the use of structured clinical pathways, protocolized alert handling, and pharmacological escalation schemes. Large-scale, prospective evaluations of the Triage-HF algorithm in a real-world clinical setting are still lacking. The INTERVENE-HF study introduced a Triage-HF-based management protocol, paving the way for further studies to investigate whether a Triage-HF-based carepath can, in fact, reduce HF hospitalizations in a real-world setting [27]. At our institution, Triage-HF is integrated into clinical practice, and Triage-HF risk score transmissions from ambulatory CIED patients are reviewed routinely in accordance with a structured carepath.

To better comprehend Triage-HF’s value and advance optimal clinical implementation, studies that evaluate the clinical efficacy of Triage-HF-based care are warranted. This study aimed to investigate the performance of a Triage-HF guided carepath in a contemporary real-world ambulatory HF patient setting, assessing its accuracy and efficacy in daily clinical practice. Accordingly, the positive and negative predictive values, the sensitivity and specificity, and the unexplained alert rate were analyzed. 

## 2. Materials and Methods

### 2.1. Study Population

In this prospective observational cohort study, all consecutive adult HF patients under follow-up at the Leiden University Medical Center (Leiden, the Netherlands) with a CIED and an activated Triage-HF algorithm were prospectively enrolled in a Triage-HF alert-guided carepath. The Triage-HF algorithm is compatible with a range of CIEDs, including implantable cardioverter-defibrillators (ICDs), cardiac resynchronization therapy pacemakers (CRT-Ps) and defibrillators (CRT-Ds), and pacemakers, that have the OptiVol feature enabled. The study period lasted from the 1 January 2023 till the 7 February 2024. Patients with a left ventricular assist device (LVAD) or complex congenital heart disease and patients with disconnected home-monitoring devices or an unwillingness to comply with the Triage-HF carepath were excluded from the analysis. 

### 2.2. Triage-HF Risk Score

As previously described in earlier studies, the Triage-HF algorithm is driven by a Bayesian belief network (BBN) [12,21,22]. The BBN is used to integrate a range of diagnostic sensor-derived parameters, including the OptiVol index based on thoracic impedance, patient activity, night heart rate (NHR), heart rate variability (HRV), atrial tachycardia/atrial fibrillation (AT/AF) burden, ventricular rate during AT/AF (VRAF), percent CRT pacing, and detected arrhythmia episodes/therapy delivered (Appendix A). Before being subjected to the BBN model, each parameter is stratified into levels, wherein lower levels indicate normal values and higher levels indicate increasingly abnormal values. The BBN is then imputed as a joint probability distribution combining these levels derived from the diagnostic parameters to compute a numeric score ranging from zero to one. This calculated probability is, in turn, translated into a monthly evaluation of the patient’s risk status. A score of less than 0.054 is categorized as low-risk, a score of 0.054–0.20 is categorized as medium-risk, and a score of 0.20 or higher is categorized as high-risk. A Triage-HF risk score management report is illustrated in Figure 1. Firstly, the monthly risk status for the prediction of an HF event in the next 30 days is presented. The following section illustrates trends for each device diagnostic parameter contributing to the risk score. The upper trend visualizes the daily risk status from the preceding 30 days, which, in turn, is used for the calculation of the monthly risk status. 

### 2.3. Triage-HF Alert-Guided Carepath

Device data, including the Triage-HF risk scores, were collected via the Carelink Platform (Medtronic, Minneapolis, MN, USA). Participants underwent scheduled monthly transmissions and were monitored in accordance with the hospital’s Triage-HF-driven carepath (Figure 2). At each transmission, the Triage-HF algorithm stratified patients as being at low, medium, or high risk for HF events in the next 30 days. Dedicated device technicians systematically reviewed all incoming high-risk transmissions, evaluating for and addressing any technical device for lead-related issues and/or identifying arrhythmias. Subsequently, the Triage-HF high-risk alerts and the CIED-specific information were forwarded to a specialized HF nurse, who contacted the patient by phone. In line with the current literature, in this current carepath, only high-risk scores were deemed actionable, due to the higher specificity and likelihood of HF-related events compared with low- and medium-risk scores. The assessment included standardized screening questions for signs and symptoms of impending fluid retention (e.g., progressive shortness of breath or fatigue, worsening peripheral edema, or weight gain indicative of fluid overload (Appendix A)). If patients with a high-risk alert had two or more signs of impending fluid retention based on the structured HF questionnaire, the alert was considered to be a true positive.

Interventions following the confirmation of impending fluid retention were based on the current ESC HF guidelines [3]. Depending on the patient’s symptoms, the underlying trigger, and the severity of fluid retention, further interventions varied from reinforcing lifestyle advice to the escalation of diuretics in (1) an ambulatory setting (oral), (2) a single-day admission for intravenous diuretics, or (3) an HF-related hospitalization setting if previously undertaken measures were deemed insufficient. If patients were symptomatic due to persistent atrial arrhythmias, an elective overpacing attempt and/or cardioversion was planned. Following an intervention, the effect was evaluated after 72 h. In the absence of signs of impending fluid retention at the initial phone contact, subsequent assessments were scheduled at 2, 6, and 10 weeks after the initial alert registration, or until transitioning to a low- or medium-risk alert. If a clinically meaningful, yet not primarily HF-related, diagnosis was suspected, the patient was referred to the general practitioner for further investigation. 

### 2.4. Data Sources and Collection

Demographic and clinical data were collected from the electronic hospital records (HiX Chipsoft Amsterdam, the Netherlands and EPD-Vision Leiden, The Netherlands). The information extracted included, among others, age, sex, type of CIED, etiology of heart failure, left ventricular ejection fraction, co-morbidities, and medication. 

### 2.5. Alert Definition

The assessment of patients following a high-risk alert was conducted systematically in accordance with the triage-HF alert-based carepath. A high-risk alert was considered to be a true positive if at least two criteria were met according to the HF questionnaire or a relevant and actionable medical problem was identified during the alert-triggered follow-up [28,29]. These clinically relevant, yet not primarily HF-related, episodes included respiratory and/or hemodynamic medical problems, which often cause secondary fluid overload, such as pulmonary infection or anemia, requiring transfusion. A high-risk alert was adjudicated as a false positive if no signs or symptoms of fluid retention were revealed during the follow-up period. An episode of congestion despite the presence of a high-risk alert in the preceding 30 days was adjudicated as a false negative. Patients were considered true negatives in the absence of a high-risk alert and the absence of signs and symptoms of decompensated HF during the follow-up window.

### 2.6. Outcome Measures

The primary outcome of this study was the diagnostic accuracy of the Triage-HF algorithm for identifying impending fluid retention. Key metrics such as sensitivity, specificity, positive predictive value, negative predictive value, and unexplained alert rate (UAR) were assessed. The secondary outcomes focused on the practicality and functionality of the Triage-HF carepath and included the alert handling time (i.e., the number of days between the transmission of an alert and first patient contact), the moment of classification as a true positive (the timeframe in which a case was identified as a true positive), sensory parameters contributing to the high-risk score status, and the interventions after an alert. 

### 2.7. Statistical Analysis

Data were analyzed using SPSS version 29 (IBM Corp, Armonk, NY, USA). Data with a normal distribution are presented as means ± standard deviation (SD), while non-normally distributed data are presented as medians with the interquartile range [IQR1–IQR3] unless otherwise specified. Normality was assessed by the visual confirmation of a Bell curve and the use of the Kolmogorov–Smirnov and Shapiro–Wilk tests. Logistic regression with a random effects model to account for repeated observations within the same patient was used to assess the accuracy of the Triage-HF high-risk scores. The sensitivity, specificity, and predictive values were determined by means of logistic regression with generalized linear mixed models. Descriptive statistics were applied to determine the handling time, the follow-up moment at which a high-risk alert was identified as a true positive, and the percentage of alerts triggered by specific sensor data. A *p*-value of <0.05 was considered statistically significant.

### 2.8. Ethics Statement

All tests and procedures were performed as part of standard clinical care. This study was conducted in accordance with the ethical standards of the institutional and/or national research committees and with the 2013 Helsinki Declaration or comparable ethical standards. Appropriate approval and a waiver for written informed consent were obtained from the institutional medical ethical board and the clinical governance division of the participating center (study protocol (2024-025)).

## 3. Results

### 3.1. Study Population

At the time of enrollment, a total of 102 patients with a CIED with the Triage-HF algorithm enabled were monitored according to the Triage-HF alert-triggered HF carepath and screened for inclusion. Ten patients were excluded from the analysis: three had an LVAD, one had a complex congenital heart disease, and six withdrew from the structured carepath due to personal preference and/or an inability to comply with the remote evaluations (Figure 3). Consequently, 92 patients underwent follow-up and were included in the current analysis. As shown in Table 1, the median age was 69 years [IQR 59–75], and 72 patients (78%) were male. A total of 60 patients (65%) were in NYHA functional class II and 14 (15%) in NYHA class III or IV. The etiology of HF was ischemic in 37 patients (40%), and the mean left ventricular ejection fraction was 38 ± 10%. A cardiac resynchronization therapy device was present in 62 patients (67%), while the remaining 30 patients (33%) had a single- or double-chamber device. The majority of patients were on a pharmacological regimen in line with current ESC guidelines for the treatment of chronic heart failure [3]. Specifically, beta-blockers were used by 92%, ACE inhibitors/ARBs/ARNIs by 91%, MRAs by 64%, SGLT2 inhibitors by 32%, and diuretics by 69%. The median follow-up duration was 9 months [IQR 5–12] and entailed a total of 61.6 patient years. None of the patients died during follow-up. 

### 3.2. Triage-HF Alerts 

During the 61.6 patient years follow-up, 36 high-risk Triage-HF alerts occurred in 26 patients. The majority of these 26 patients experienced a single high-risk alert episode (n = 19 (73%)), while the other 7 patients (27%) had two or more high-risk alert episodes. The remaining 66 patients did not experience any high-risk alert during follow-up. On average, there were 0.58 high-risk alert episodes per patient year.

### 3.3. Clinical Performance of the Triage-HF Alerts

Based on the previously defined criteria, 31 of the 36 high-risk alerts (86%) were identified as true positives for impending fluid retention (Figure 4). Of interest, 5 of these 31 true positive alerts were clinically relevant but not primarily *HF*-driven (e.g., congestion triggered by infection, anemia, etc.). In total, 5 of the 36 high-risk alerts (14%) were classified as false positives. Two of these five alert episodes were triggered by asymptomatic atrial fibrillation or flutter, while in the remaining three alert episodes, no sensible explanation/trigger could be identified. As a result, the unexplained alert rate (UAR) was calculated at 0.05 alerts per patient year. There were seven episodes in five patients during which they experienced signs and symptoms of fluid retention but did not have a high-risk alert in the preceding 30 days, thus being classified as false negatives. In three of these episodes, the patients were eventually hospitalized for *HF*. The false negative rate was 0.11 events per patient year.

As shown in Table 2, based on these 36 high-risk alerts, logistic regression with generalized linear mixed models estimated the sensitivity of the Triage-HF algorithm-based carepath for detecting early signs of fluid retention at 83% (CI 65–92%). The estimated specificity was 97% (CI 92–99%). Furthermore, the estimated positive predictive value was 89% (CI 73–96%), and the estimated negative predictive value was 94% (CI 89–97%). 

### 3.4. Alert Handling and Impact on Patient Management

The median alert handling time was 2 days [IQR 1–9]. Among the 26 high-risk alert episodes that were primarily HF-related, 18 high-risk alert episodes (69%) involved the reinforcement of lifestyle advice and the intensification of oral treatment in the home setting. In two high-risk alert episodes (8%) intravenous administration of diuretics was necessary at the day clinic, and in six high-risk alert episodes (23%), an HF hospitalization was inevitable. 

### 3.5. Timing of True Positive Alert Adjudication

Among the 31 high-risk alert episodes adjudicated as true positives, 18 (58%) resulted in manifestations of signs and symptoms of impending fluid retention within the first week of follow-up (Figure 5). Subsequently, nine high-risk alerts (29%) were classified as true positives in the second–third weeks of follow-up, while the remaining four high-risk alerts (13%) were classified as true positives between the fourth and sixth weeks of follow-up. 

### 3.6. Device-Specific Diagnostic Parameters Contributing to a Triage-HF High-Risk Status

Figure 6 displays the most prevalent sensor deviations that contributed to the high-risk alerts. Specifically, an elevated night ventricular rate (84%; n = 26), an OptiVol alert (71%; n = 22), and reduced patent activity (71%; n = 22) were most frequently encountered.

## 4. Discussion

The main finding of this study was that the CIED-based Triage-HF algorithm embedded in a clinical carepath was robust in detecting impeding fluid retention in a real-world ambulatory HF setting. In particular, the demonstrated sensitivity was 83%, the specificity was 97%, the positive predictive value was 89%, and the negative predictive value was 94%. The unexplained alert rate was rather low, with 0.05 alerts per patient year. 

The CIED-based Triage-HF algorithm is exemplary of a contemporary multisensory algorithm and has been advocated as a reliable integration for device-based monitoring [27]. Earlier studies reported on Triage-HF algorithm-based carepaths’ efficacy in managing ambulatory HF patients and identifying clinically relevant changes in patients’ well-being [23]. However, these studies reported a wide variability in the sensitivity, specificity, and positive predictive values of the Triage-HF algorithm for predicting HF hospitalizations [12,22,23,24,25,26]. For example, Okumura et al. and Burri et al. (post hoc analyses) both reported a lower sensitivity of 31.5% and 37.4%, respectively, compared with the sensitivity of 83% obtained in the current study [22,24], and the specificity was 89.0% and 90.1%, respectively, compared with the 97% reported in the current cohort. This variability in the performance of the same algorithm can, at least partly, be explained by marked differences in the definitions of HF-related events and variations in the use of structured clinical pathways, protocolized alert handling, and pharmacological escalation schemes. For example, in the study by Burri et al., an HF event comprised HF-related hospitalizations, while Okumura et al. defined HF-related events as HF-related hospitalizations and outpatient visit clinics with documented intervention. In the current study, a high-risk alert was considered a true positive if at least two criteria were met according to the previously validated HF questionnaire or if a relevant and actionable medical problem was identified, potentially resulting in higher sensitivity and specificity [29,30,31]. From a methodological perspective, the current study provides a realistic reflection of daily clinical practice and takes into account the phenomenon of “repeated measures” for the calculation of the predictive value. To evaluate the algorithm’s diagnostic accuracy, logistic regression with generalized linear mixed models was used, rather than a conventional ‘2 by 2 table’ approach. 

In the study by Ahmed et al. with similar outcome measures, a sensitivity of 98.6%, and a specificity of 63.4% were reported [23]. In the present HF carepath, it was mandated by protocol for transmissions to be scheduled monthly and for a high-risk alert to result in contact between the HF team and the patient. Interestingly, this approach led to higher predictive markers than the previous study by Ahmed et al. with a similar carepath [23]. A possible explanation for this difference could be their use of a 3-monthly transmission schedule, potentially causing alerts to lag behind the actual clinical status of the patient and thus reflecting a higher chance of missing impending fluid retention. These variations reflect the impact of the frequency of transmissions on the predictive value of the algorithm and might explain why it may be more or less efficient in a specific setting. It naturally always involves the weighting of the ‘work burden’ on the medical staff who process the transmissions and the ‘win’ of preventing a heart failure-related admission. Earlier studies raised concerns about the increased workload remote monitoring brings and found no benefits in weekly scheduled transmissions compared with standard care in terms of mortality or unplanned cardiovascular hospitalization [16,32]. However, the current carepath addresses this concern by focusing only on high-risk alerts, organizing the care into a high-tech, low-labor model requiring only active outreach to those patients deemed at the highest risk of worsening heart failure. 

Of interest, this study demonstrated the significant contributions of night ventricular rate, patient activity, and OptiVol sensor data in triggering high-risk alerts. While these parameters offer insight, a more comprehensive impression of patients’ health likely requires additional and individualized information on the etiology and stage of heart failure, (neurohumoral) biomarkers, and comorbidities and likely warrants further investigation. 

Although HF hospitalization was relatively prevalent (23%) in this study, the interpretation of this should be careful as there was no control arm, and only the patients with high-risk alerts were addressed in this carepath. In this light, the frequency of scheduled transmissions requires further (comparative) evaluation. It could be suggested that only selecting the ‘high alerts’ at monthly evaluations resulted in patients in relatively advanced stages of congestion being approached, limiting the effect that can be attained with initial ambulatory diuretic adjustments. This scenario stresses the tricky trade-off between the frequency of transmissions, the associated workload, and the sensitivity of alert-based carepaths. In contrast, the current relatively low unexplained alert rate of 0.05 alerts per patient year can probably (in part) be attributed to the less frequent monitoring intensity. This raises the question of whether real-time transmissions in future generations of CIEDs and automated alert processing hubs can efficiently and effectively manage larger patient cohorts. Due to the observational single-arm study design, the potential reduction in hospitalization remains uncertain, making it difficult to conduct a cost–benefit comparison between the burden on the healthcare system and the effectiveness of implementing the algorithm-based carepath. The resources required for frequent monitoring need to be justified by evidence of improved patient outcomes and reduced overall healthcare costs. Further studies should, therefore, also focus on the economic impact and care consumption to ensure the sustainability of implementing such monitoring systems on a larger scale.

## 5. Limitations

The findings of this study should be interpreted in the context of its open-label and observational design. The awareness of monitoring possibilities among patients and healthcare providers may have introduced selection and reporting bias. The study cohort was relatively small and single-center in nature; however, the demographic and clinical patient characteristics are representative of the broader HF patient group with a CIED. The tertiary care institution involved had substantial experience in the rhythm device-based remote monitoring of (HF) patients, and the results attained with the current carepath should, therefore, be validated in other (regional) hospital settings.

## 6. Conclusions 

The CIED-based Triage-HF algorithm embedded in a structured clinical carepath shows promise in the timely detection of impending fluid retention, allowing for effective lifestyle and pharmacological interventions to prevent further deterioration that may otherwise result in a heart failure-related admission. The current findings justify further multicenter prospective and randomized studies to evaluate the clinical impact on HF-related hospital admissions and survival. 

## Figures and Tables

**Figure 1 sensors-24-03664-f001:**
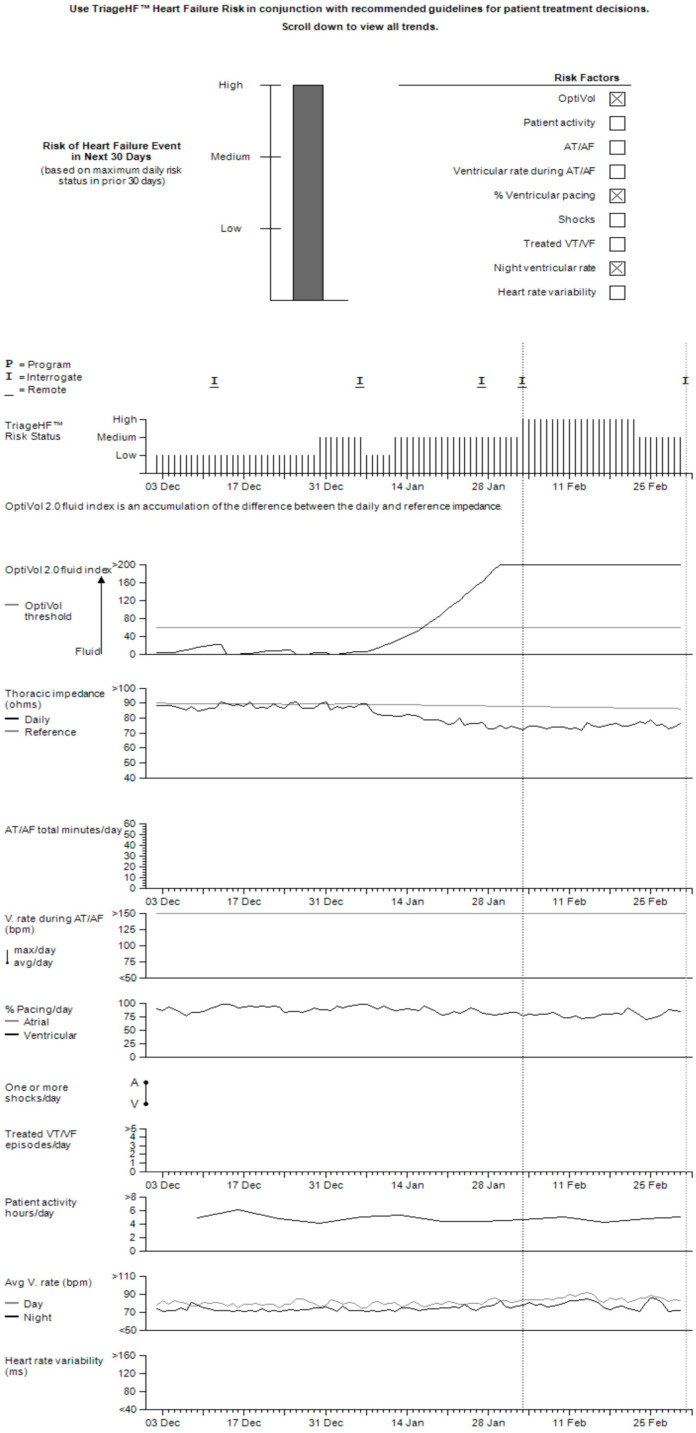
Heart failure management status report. A heart failure management status report from CareLinkTM (Medtronick, Minneapolis, MN, USA). The top component of the report shows the future 30-day risk for a patient at high risk, including device parameters that contribute to that risk. The first trend displays the daily risk status, which is dynamic and used to derive the future 30-day risk status. The trends below are specific for each device parameter.

**Figure 2 sensors-24-03664-f002:**
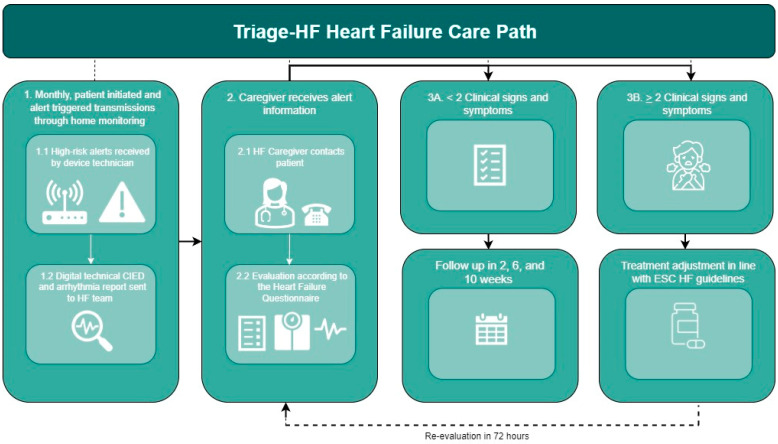
Triage-HF Carepath. CIED, Cardiac Implantable Electronic Device; HF, heart failure; ESC, European Society of Cardiology.

**Figure 3 sensors-24-03664-f003:**
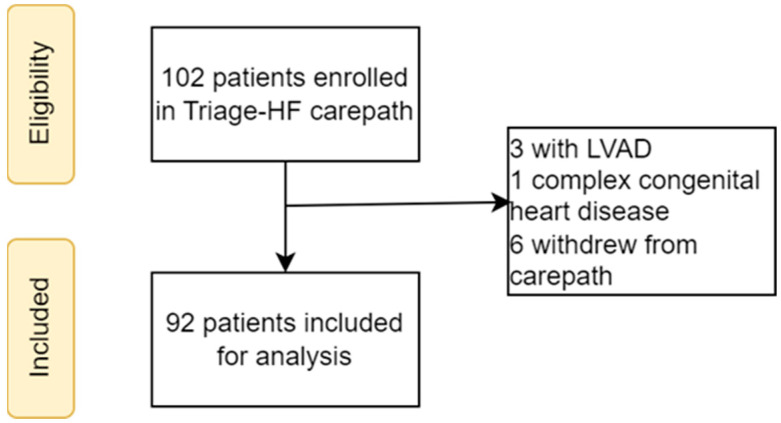
Flowchart of the Study Cohort. LVAD, left ventricular assist device.

**Figure 4 sensors-24-03664-f004:**
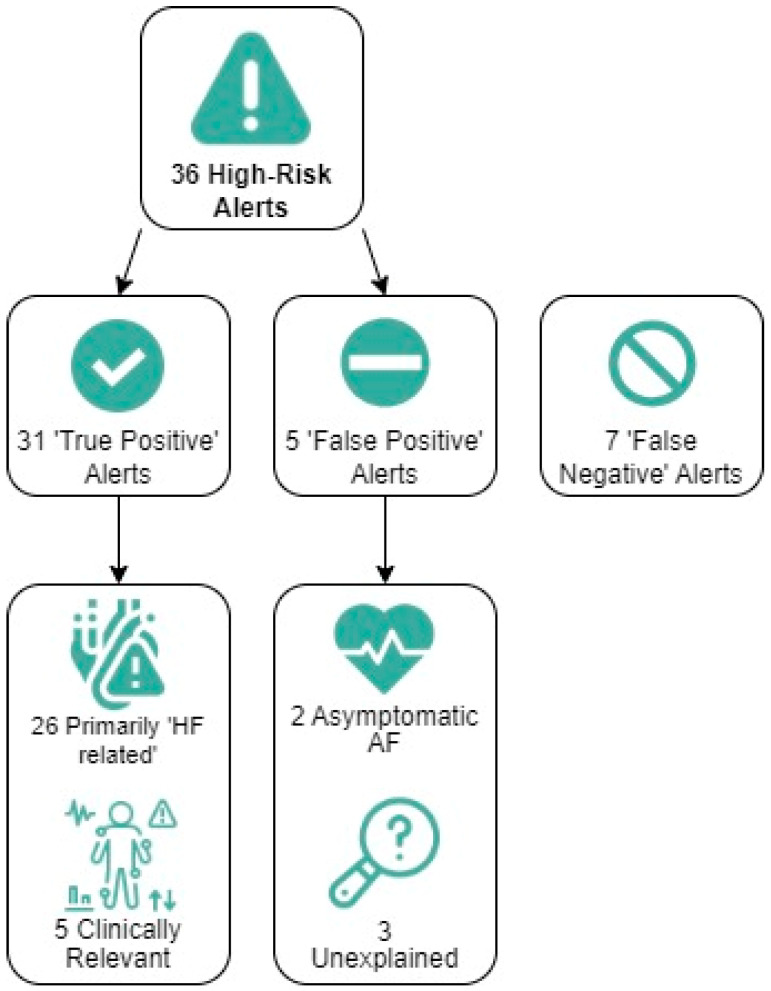
Assessment of High-Hisk Triage-HF Alerts. AF, atrial fibrillation; HF, heart failure.

**Figure 5 sensors-24-03664-f005:**
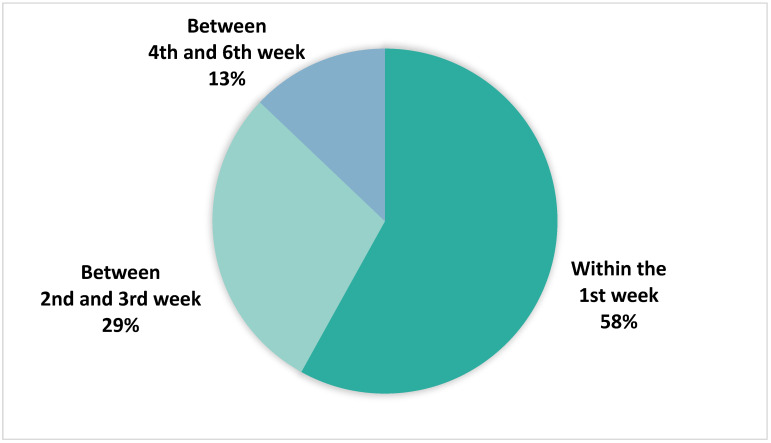
Follow-up Moment in Weeks Where a High-Risk Alert was Adjudicated as a True Positive.

**Figure 6 sensors-24-03664-f006:**
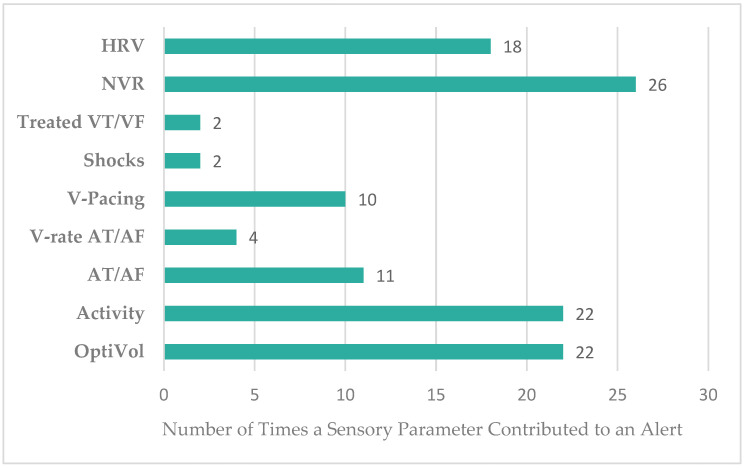
Sensory Triggers that Contributed to a True Positive High-Risk Alert. HRV, heart rate variability; NVR, night ventricular rate; Treated VT/VF, treated ventricular tachycardia/ventricular fibrillation; V-pacing, ventricular pacing; V-rate AT/AF, ventricular rate atrial tachycardia/atrial fibrillation.

**Table 1 sensors-24-03664-t001:** Baseline Characteristics of the Patients Included for Analysis.

	n = 92
Age in years, median [IQR]	69 [59–75]
Males, n (%)	72 (78%)
Years since HF diagnosis, median [IQR]	10 [6–17]
BMI (kg/m^2^), median [IQR]	26 [24–30]
**LVEF in % (SD)**	38 ± 10
**NYHA class, n (%)**	
I	18 (20%)
II	60 (65%)
III–IV	14 (15%)
**Etiology**	
Ischemic	37 (40%)
Non-ischemic	55 (60%)
**Device**	
CRT, n (%)	62 (67%)
Percentage of biv-pacing (%), median [IQR]	99 [95–100]
DDD/VVI ICD	30 (33%)
**Cardiac history**	
CABG, n (%)	16 (17%)
Valve surgery, n (%)	18 (20%)
**Co-morbidities**	
Atrial fibrillation, n (%)	37 (40%)
Hypertension, n (%)	24 (26%)
Diabetes mellitus, n (%)	23 (25%)
Ischemic CVA/TIA, n (%)	16 (17%)
**Laboratory findings**	
NT-proBNP in ng/L, median [IQR]	512 [165–1668]
Hb in mmol/L, median [IQR]	8.7 [8.1–9.4]
eGFR in mL/min/1.73 m^2^, median [IQR]	66 [46–83]
**Pharmacotherapy**	
Beta-blockers, n (%)	88 (92%)
ACE/ARB/ARNI, n (%)	84 (91%)
MRA, n (%)	59 (64%)
SGLT2i, n (%)	29 (32%)
Diuretics, n (%)	63 (69%)

n, number of patients; IQR, inter-quartile range; SD, standard deviation; HF, heart failure; BMI, body mass index; NYHA, New York Heart Association functional class; LVEF, left ventricular ejection fraction; CRT, cardiac resynchronization therapy; biv, biventricular; CABG, coronary artery bypass graft; CVA, cerebral vascular accident; TIA, transient ischemic attack; NT-proBNP, N-terminal pro-B-type natriuretic peptide, normal levels < 247.0 ng/L; Hb, hemoglobin, normal levels 7.5–10 mmol/L in females and 8.5–11 mmol/L in males; eGFR, effective glomerular filtration rate, normal levels > 60.0 mL/min/1.73 m^2^; ACE, angiotensin-converting enzyme; ARB, angiotensin II receptor blocker; ARNI, angiotensin receptor neprilysin inhibitor; MRA, mineralocorticoid receptor antagonist; SGLT2i, sodium–glucose cotransporter-2 inhibitor.

**Table 2 sensors-24-03664-t002:** Accuracy of the Triage-HF Algorithm.

	Population Mean	95% Confidence Interval
**Sensitivity**	83%	65–92%
**Specificity**	97%	92–99%
**Positive predictive value**	89%	73–96%
**Negative predictive value**	94%	89–97%
**Unexplained alert rate**	0.05 alerts per patient year	

## Data Availability

The data presented in this study are available upon reasonable request through the corresponding author. The data are not publicly available due to privacy restrictions.

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
