# Peer review of "Efficacy of the Cardiac Implantable Electronic Device Multisensory Triage-HF Algorithm in Heart Failure Care: A Real-World Clinical Experience"

_sensors, 2024, doi:10.3390/s24113664_

Round 1

Reviewer 1 Report

Comments and Suggestions for Authors

Please enhance the last paragraph of the introduction to underscore the rationale of this study.

Sensitivity, specificity, and PPV exhibited considerable variation across different studies, a phenomenon noted by the authors. They attributed this wide variability in part to discrepancies in the definition of heart failure events across studies. It is pertinent to elucidate the author’s approach to mitigating this variability, such as employing high-risk alert patient analysis exclusively, and to delineate the study’s hypotheses.

As acknowledged by the authors, the cohort of subjects identified with high-risk alerts was limited and sourced from a single center. It is advisable to augment the sample size or to stratify and analyze the outcomes of studies involving low- and medium-risk alert subjects separately, as well as collectively.

The authors stated that a high-risk alert was considered a true positive if at least two criteria were met according to the HF questionnaire or if a relevant and actionable medical problem was identified during follow-up from the alert. Are the criteria for identifying high-risk alert patients consistent and do subjects have similar symptoms?

The data outlined in the ‘Timing of True Positive Alert Adjudication’ section deviates from that provided in the Abstract.

How is the unexplained alert rate calculated?

Provide a succinct overview of the criteria delineating low, medium, and high-risk alert HF events.

Consider modifying the background color of Figure 1 to enhance readability and highlight text and images.

Table 2: There appears to be a significant proportion of patients concurrently using multiple medications. Please explain this.

While statistical methodologies were employed, the statistically analyzed findings are not clearly presented in the Results and Discussion section. Please elaborate on which data underwent statistical analysis and elucidate the corresponding results.

Reviewer 2 Report

Comments and Suggestions for Authors

In this paper, the authors present results relating to assessment of the diagnostic accuracy of the Triage-HF algorithm (Medtronic) for identifying impending fluid retention, using multi-parametric sensor analysis. As key metrics are used statistical parameters - sensitivity, specificity, positive predictive value, negative predictive value and unexplained alert rate. The clinical trial included 92 adult patients with diagnosed heart failure and with the Triage-HF algorithm activated of their cardiac implantable electronic device (CIED). The duration of the study is about 13 months. Additional parameters as alert handling time, the moment of classification as true positive, sensory parameters contributing to the high-risk score status and the interventions after an alert, are also analized to estimate the practicality and functionality of the Triage-HF care path.

 The paper is well structured. The abstract correctly present the content of the article. I evaluate the methodology as adequate for the purposes of the study. Reference sources are relevant to the content and are cited at appropriate places in the text.

The authors correctly note certain shortcomings related to the relatively small group of patients, as well as possible bias.

My recommendation is to move two paragraphs from the discussion section (line 279-290 and line 293-306) to the introduction section.

Reviewer 3 Report

Comments and Suggestions for Authors

Review 

In this compelling paper, the authors assessed the accuracy of a real-world heart failure triage protocol based on a cardiac device heart failure algorithm, a topic of considerable importance in clinical practice. The paper exhibits a well-structured and articulate presentation. The primary objective was to ascertain the diagnostic accuracy of the Triage-HF algorithm in detecting fluid retention. The Triage carepath algorithm is presented in a clear and easily understandable format.

Both the introduction and discussion sections are robustly organized.

Here are my main suggestions for improvement:

  1. It remains unclear how high/moderate/low risk alerts were defined, which is a fundamental aspect of this study. This information should be included in the methods section, potentially through the use of a table or incorporation into Figure 1.
  2. The specific HF signs utilized in the HF algorithm should be detailed, either in a table with explanations for the device alerts or as part of Figure 1.
  3. Was there a statistical rationale for enrolling 92 patients? If this was done for convenience, it should be explicitly stated.
  4. Clarification is needed regarding the inclusion criteria for this study, particularly why pacemaker patients were not included, given that all patients had either ICDs or CRT.
  5. The study focused on a high-risk population, with most patients already symptomatic at evaluation with ICDs or CRT. This raises questions about whether such a population might influence the evaluation of the protocol's accuracy, potentially impacting positive and negative predictive values.
  6. Despite 40% of patients having afib, there were relatively few UARs. How was the  antiarrhythmics use? Did they have prior Afib ablation, or AV node ablation?
  7. Another important consideration is the burden on the healthcare system and cost-effectiveness of this evaluation. If the potential reduction in hospitalizations is uncertain, why should healthcare systems implement it, even for "high-risk alarms"?
  8. In the conclusion, it may be prudent to refrain from stating definitively that the carepath can effectively and timely... As acknowledged in the limitations, this is an observational study. A conclusion suggesting that the algorithm shows promise and merits further research would align better with the discussion.
  9. Minor point: Please include normal NT-BNP levels for your institution.

By addressing these points, the paper could further enhance its clarity and impact.
